# The Nucleocapsid of Paramyxoviruses: Structure and Function of an Encapsidated Template

**DOI:** 10.3390/v13122465

**Published:** 2021-12-09

**Authors:** Louis-Marie Bloyet

**Affiliations:** Department of Molecular Microbiology, School of Medicine, Washington University in Saint Louis, St. Louis, MO 63110, USA; louis-marie.bloyet@wustl.edu

**Keywords:** paramyxoviruses, nucleocapsid, nucleoprotein, phosphoprotein, polymerase complex, intrinsic disorder

## Abstract

Viruses of the *Paramyxoviridae* family share a common and complex molecular machinery for transcribing and replicating their genomes. Their non-segmented, negative-strand RNA genome is encased in a tight homopolymer of viral nucleoproteins (N). This ribonucleoprotein complex, termed a nucleocapsid, is the template of the viral polymerase complex made of the large protein (L) and its co-factor, the phosphoprotein (P). This review summarizes the current knowledge on several aspects of paramyxovirus transcription and replication, including structural and functional data on (1) the architecture of the nucleocapsid (structure of the nucleoprotein, interprotomer contacts, interaction with RNA, and organization of the disordered C-terminal tail of N), (2) the encapsidation of the genomic RNAs (structure of the nucleoprotein in complex with its chaperon P and kinetics of RNA encapsidation in vitro), and (3) the use of the nucleocapsid as a template for the polymerase complex (release of the encased RNA and interaction network allowing the progress of the polymerase complex). Finally, this review presents models of paramyxovirus transcription and replication.

## 1. Introduction

Members of the *Paramyxoviridae* family are enveloped viruses with non-segmented negative-strand RNA genomes. These viruses share similar gene expression and genome replication mechanisms to the members of the other ten families of the *Mononegavirales* order. Paramyxoviruses are currently divided into 4 subfamilies, 17 genera, and 78 species (Figure 1) [1] (information as of November 2021, for an update see https://talk.ictvonline.org/taxonomy/). Paramyxoviruses are found in a broad range of animals, including mammals, fishes, reptiles, or birds and include several human and animal pathogens such as measles virus (MeV), mumps virus (MuV), canine distemper virus (CDV), and Newcastle disease virus (NDV). Some, such as Nipah virus (NiV) and Hendra virus (HeV), regularly emerge in human or animal populations where they cause deadly epidemics. Although vaccines exist against some paramyxoviruses, effective antiviral treatments for post-exposure prophylaxis are urgently needed [2,3].

Viral transcription and replication are both mediated by an RNA synthesis machinery composed of a viral polymerase complex and its encapsidated template. The polymerase complex consists of the large protein (L) and its co-factor the phosphoprotein (P) [4,5,6,7]. This complex uses a genome (or antigenome) embedded into a homopolymer of viral nucleoproteins (N) as a template [8,9,10]. The encapsidation of the RNA in this ribonucleoprotein structure, known as the nucleocapsid, prevents the annealing of positive and negative RNA strands into double-stranded RNA structures, as well as the folding of genomic RNA into secondary RNA structures, which helps to prevent recognition by innate immune receptors [11,12]. It also protects the genome from degradation by nucleases [13,14] and from being targeted by siRNA [15,16].

The mechanisms used by the polymerase complex to access the encased RNA, progress along the nucleocapsid, and generate new encapsidated products with soluble nucleoprotein monomers are yet to be fully deciphered. However, in the past few years, the atomic structures of the nucleoprotein of a number of paramyxoviruses from different genera have been solved in either assembled [17,18,19,20,21,22,23] or non-assembled form [24,25,26,27] (Figure 1). Moreover, since the atomic structure of the polymerase complex of parainfluenza virus 5 (PIV5) has been solved, structural information is now available for all the components of the RNA synthesis machinery of paramyxoviruses [28]. Significant progress has also been made on the use of the nucleocapsid as a template, especially on the complex interaction network at play between the nucleoprotein and the phosphoprotein. This review summarizes both structural and functional insights on the nucleocapsid of paramyxoviruses, its assembly, and its role as a template. Models of transcription and replication are proposed.

## 2. Gene Expression and Genome Replication

### 2.1. The Viral Genome

The genomes of paramyxoviruses contain a single, bipartite, transcription promoter at the 3′ end, followed by at least six conserved, adjacent genes separated by non-transcribed intergenic regions (Figure 2A) [29,30,31]. The lengths of the genomes range from 14 to 21 kb and must be multiples of six, the so-called ‘rule of six’ [32,33]. The conserved genes encode for the nucleoprotein, the phosphoprotein, the matrix protein, the fusion protein, the receptor-binding protein, and the large protein, or polymerase (Figure 2A). While the N, P, and L proteins are the essential components of the viral RNA synthesis machinery, the matrix protein and the two glycoproteins are necessary for the formation of the viral particles and their entry into cells [31].

### 2.2. Components of the RNA Synthesis Machinery

The nucleoproteins of paramyxoviruses are made of a folded domain (N_core_) followed by a long intrinsically disordered tail (N_tail_) [34] (Figure 2B). While N_core_ is responsible for the oligomerization on the RNA and constitutes the main body of the nucleocapsid [35,36,37,38], N_tail_ mediates the recruitment of the polymerase complex [34,39,40]. The nucleoproteins self-assemble on the genome due to their affinity for RNA and the interactions between adjacent protomers, as discussed later.

The L protein is a large, multi-domain protein, made of five globular domains linked together by flexible linkers [28,41] (for a review see [42,43]) (Figure 2B). It carries all the enzymatic activities required for transcribing and replicating the genome: RNA synthesis [5,6], mRNA cap addition [44,45], guanine-N-7-methylation of the cap, and ribose 2′O-methylation of the first nucleotide [46,47]. Polyadenylation of the mRNA occurs when the polymerase stutters on a short poly(U) sequence located in a polyadenylation signal [48,49].

The phosphoprotein plays several essential roles: it stabilizes the L protein [7,50,51,52], recruits the L protein on its template [53], stabilizes the N protein in a monomeric and RNA-free state [54,55,56], and provides N proteins for the encapsidation of nascent RNA during replication [7,55]. The phosphoprotein’s sequence is poorly conserved but its organization in three modules linked by long disordered regions is shared within the viral order [57,58,59,60,61,62] (Figure 2B). The N-terminal domain (PNTD) binds the nucleoprotein to prevent its self-assembly on RNAs other than viral genomes and antigenomes [55,56]. This interaction forms the N0-P complex, used as a substrate for the encapsidation of the nascent RNA during genome replication [7]. The central domain, or oligomerization domain (POD), forms a long, parallel, coiled-coil tetramer for Sendai virus (SeV) [63], MeV [64,65], NiV [62,66], PIV5 [28], and human parainfluenza virus 3 (hPIV3) [67], but an antiparallel dimer of two parallel coiled-coil dimers for MuV [68,69]. The C-terminal domain, or X domain (PXD), interacts with the nucleoprotein and recruits the polymerase on the nucleocapsid [50,70,71,72,73,74,75].

### 2.3. Transcription and Replication

Following fusion of viral and host cell membranes during entry, the polymerase complexes packaged in virions transcribe the viral genes into capped and polyadenylated mRNAs (Figure 2C) [31]. The viral genes are separated by short non-transcribed intergenic regions and contain a gene start (GS) and a gene end (GE) signal at their 3′ and 5′ ends, respectively (Figure 2A). The current model suggests that RNA synthesis starts at the 3′ end of the genome where the polymerase recognizes the transcription promoter. The L protein first synthesizes the leader RNA, a short, uncapped RNA, which is released upon recognition of the first intergenic region. The polymerase then scans the genome to find the first GS signal and re-initiates RNA synthesis. The nascent RNA is capped and methylated by the L protein. Upon recognition of the GE signal, the polymerase adds a poly(A) tail, releases the mRNA, and scans the genome to find the next gene start signal. As the transcription re-initiation is not perfectly efficient, the polymerase generates a decreasing gradient of mRNAs (Figure 2C).

Once the concentration of nucleoproteins reaches a certain threshold, the polymerase can switch from a transcription mode to a replication mode [76,77]. Although the accumulation of N proteins may not be the only trigger, artificially increasing the amount of N proteins advances the onset of replication [77,78]. Initiation of replication occurs when the nascent leader RNA is encapsidated by monomers of N and the polymerase ignores the transcription signals, instead synthesizing a full-length encapsidated copy of the genome (antigenome) [76,79]. This antigenome is then used as a template to generate full-length encapsidated copies of the viral genome (Figure 2C).

Viral transcription and replication take place in the cytoplasm in membraneless inclusion bodies. Viral genomes, together with N, P, and L proteins, are concentrated in these inclusions formed by liquid-liquid phase separations. The disordered regions of the N and P proteins play a critical role in the formation of these liquid-like structures (for a review, see [80,81,82]).

Finally, viral transcription and replication are also affected by the phosphorylation status of N and P. Indeed, both N_tail_ [83,84,85] and the disordered region upstream of P_OD_ [86,87,88,89,90,91,92,93,94] contain major phosphorylation sites targeted by cellular kinases [95,96,97,98,99,100,101]. Although the roles of these phosphorylations are still unclear, data suggest the phosphorylation of some residues can downregulate [85,92,93,100,102] or upregulate RNA synthesis [83,84,91], modify the balance between transcription and replication [103], alter the stability of encapsidated genomes [104], participate in RNA encapsidation [105], tune the interaction between N and P [94], or facilitate the growth of the inclusion bodies [106].

## 3. Structure of the Nucleocapsid

### 3.1. Overall Architecture

Nucleocapsids extracted from disrupted virions or infected cells form left-handed, rod-like helical structures with a herringbone appearance under an electron-microscope [107,108,109,110] (Figure 3A). The coiling and rigidity of the helix depend on the conditions of the milieu, such as the ionic strength [111]. Indeed, while, at low salt concentration, the nucleocapsids are loose and flexible, and at high ionic strength, the helixes are tight and rigid with a length of about 1 µm and a diameter of 15–20 nm. The flexibility also varies from one virus to another, with the nucleocapsids of MuV being more flexible than the nucleocapsids of MeV, themselves more flexible than the nucleocapsids of SeV or PIV5 [109,112]. Purified intact nucleocapsids from the same virus, or even sections of a single nucleocapsid, can also adopt condensed conformations with different pitches ranging from 5.3 to 6.8 nm for SeV [110], 5.2 to 6.6 nm for MeV [113], or 5.8 to 6.7 nm for MuV [114] (Figure 3C). A smaller pitch correlates with a more rigid and straight helix [112]. In intact MeV particles, the nucleocapsids have a higher pitch (7–8 nm), thus raising the question of whether a preferred pitch exists in physiological conditions and which value it adopts [115]. The average number of N subunits per helical turn is relatively conserved and varies from 12.3 to 13.4 [18,110,113,114]. Removal of the encapsidated RNA from MuV nucleocapsid increases the flexibility of the helix but does not affect its helical conformation, indicating that the nucleocapsid assembly is stable even without RNA [114].

Expression of the nucleoprotein in avian cells [118], mammalian cells [36], insect cells [119], or bacteria [120], induces the formation of nucleocapsid-like complexes (NCLC) containing cellular RNAs. These complexes can form helices [118,121], rings [17,122], or clam-shaped structures [19,21,22] (Figure 3B–D). The last has been observed for NDV, SeV, and NiV, and is made of two helical turns interacting in a back-to-back manner (Figure 3D). The detection of this structure for paramyxoviruses from different genera and its presence in nucleocapsids extracted from NDV and SeV virions, suggests it may be biologically relevant [19,21]. The clam-shaped structure has been proposed to play a role in seeding the assembly of double-headed helices, protecting the 5′ end of the genome from nucleases, and promoting the encapsidation of multiple nucleocapsids per virion [123,124,125].

### 3.2. Structure of the Nucleoprotein

The sequence and the organization of N_core_ are well conserved across the *Paramyxoviridae* family, while the sequence and length of N_tail_ vary greatly (Figure 4A). The structure of the N protein in an oligomeric, RNA-bound conformation (i.e., engaged in an NCLC) have been solved for PIV5 [17], MeV [18,20], NDV [19], cetacean morbillivirus (CeMV) [23], NiV [22], SeV [21], and MuV [126], while the structure of N in complex with P_NTD_ (i.e., N^0^-P complex) was solved for NiV [24], MeV [25], PIV5 [26], and hPIV3 [27] (Figure 1).

N_core_ consists of two globular domains, the N-terminal (N_NTD_) and C-terminal domains (N_CTD_), which form a positively charged cavity to accommodate the RNA (Figure 4B,C). These domains are preceded and followed by an N-terminal and a C-terminal arm (N_NTDarm_ and N_CTDarm_) connected by flexible linkers; these arms are responsible for the self-assembly of N protomers onto the RNA. The superposition of the structures of the N proteins of paramyxoviruses shows high conservation of the tertiary and secondary structures (Figure 4D). The global architecture is also well conserved among members of the *Mononegavirales* order, since the N proteins of respiratory syncytial virus (RSV) [129,130] and human metapneumovirus [131] (two pneumoviruses), Ebola virus [132,133,134,135,136] and Marburg virus [137] (two filoviruses), Borna disease virus [138] (a bornavirus), vesicular stomatitis Indiana virus (VSIV) [139] and rabies virus [140] (two rhabdoviruses) have similar structures and modes of self-assembly (for a review see [141]) (Figure 4E). 

### 3.3. Interactions between Protomers

Each N protomer forms tight connections with its neighbors via its two arms protruding on each side of the N protein (Figure 5A). On one side, the N_NTDarm_ of the N_i_ protomer forms an alpha helix that binds a hydrophobic groove on the back of the N_CTD_ of the N_i-1_ protomer (Figure 5B(i)). On the other side, the N_CTDarm_ folds in two alpha helixes interacting with the top of the N_CTD_ of the N_i+1_ protomer (Figure 5B(ii)). A third tight connection has been described between an extended loop of the N_NTD_ of the N_i_ protomer that enters into a hole formed by the N_NTD_ and N_NTDarm_ of the N_i+1_ protomer (Figure 5B(iii)) [21,126]. In addition to these interactions, there are large contact areas between adjacent protomers with numerous additional bonds [23].

These N-N interactions curve the N oligomer so that the RNA is exposed on the outside of the helix (or ring) and all the tight inter-protomer connections are hidden inside the helix (Figure 5C) [17,18,117]. This conformation is similar to that observed in the nucleocapsids of filoviruses [134] and pneumoviruses [130], but opposite to the one found in rhabdoviruses [142,143] where the RNA is located in the interior of the helical nucleocapsid.

### 3.4. Interactions with RNA

The genomic RNA is inserted in a cavity formed by the N_NTD_ and the N_CTD_ (Figure 6A,B). Each N protomer binds six nucleotides in agreement with the rule of six [17,18]. The RNA is mainly stabilized by interactions between positively charged residues and the RNA backbone, thus explaining the low sequence specificity of encapsidation (Figure 6A). The RNA adopts a regular conformation with a three-base stack oriented outwards to the solvent (bases “out”) and the following three-base stack facing toward the protein (bases “in”) (Figure 6B).

Using recombinant N proteins fused to a cleavable P_NTD_, NCLCs were generated in vitro after cleavage of the P_NTD_ and incubation with either a six-nucleotide-long poly(A) RNA or an RNA corresponding to the first six nucleotides of the viral genome [20,23]. For both MeV and CeMV, NCLCs adopt a helical conformation with the RNA hexamer in the same phase register. These structures suggest that the first two nucleotides of the viral genome point toward the solvent and that the 3′ end of the genome is accessible to the viral polymerase (Figure 6B). This register is in agreement with data showing that cytidines at positions 1 and 6, which are both in “out” positions, are more sensitive to chemical modifications [144].

Paramyxoviruses have bipartite promoters [145,146,147,148] (for a review see [30]). The first promoter element (PE1) contains at least the first twelve nucleotides of the genome (Figure 2A). The second promoter element (PE2) is composed of three successive hexamer sequences: 3′-CNNNNN-5′ located at positions 14 to 16 (in the hexamer register) for orthoparamyxoviruses [146], and 3′-NNNNCG-5′ at position 13 to 15 for rubulaviruses [147] and avulaviruses [148]. According to the RNA register described for MeV and the average of 13 N per turn for native nucleocapsids, PE2 is located in the successive turn of the helical nucleocapsid just under PE1 in the helical conformation of the nucleocapsid (Figure 6C). Moreover, the conserved nucleotides of the two types of PE2 are facing the solvent, thus reinforcing the idea that these bases could be sensed by the polymerase complex and participate in promoting RNA synthesis.

The side chain of the residue Q202 of the nucleoproteins of PIV5 and MeV interacts with the base of the nucleotide in position six, and thus the first nucleotide of the promoter [17,20] (Figure 6A). In a cell-based minigenome assay, mutation Q202A of the N protein of human parainfluenza virus 2 allows the transcription and replication of the minigenome, irrespectively of whether its length is a multiple of six and the presence of the PE2 element [149,150]. The authors suggest that residue Q202 may stabilize the first nucleotide of the genome, thus inhibiting the initiation of RNA synthesis by the viral polymerase. PE2 would then promote RNA synthesis initiation by stabilizing the polymerase complex on the promoter. Since the function of PE2 depends on its phase register, the requirement of PE2 would ensure the conservation of the rule of six [149,150].

In addition to its impact on the promoters, the phase of the RNA bases also affects the transcription signals. Indeed, during transcription, the P messenger RNA of paramyxoviruses can be edited by the viral polymerase which adds one or several extra G residues by stuttering along a short stretch of C residues [151]. Incremental modifications of the phase register of the editing site modify the efficiency of the editing and the number of residues added [144]. Moreover, within each genus, the gene start signals are found in a few preferred phases, suggesting the phasing of the gene start signal may also affect the efficiency of the re-initiation of transcription at gene junctions [33]. Overall, these results show that the imprint of the N protomers on the RNA influences how the viral polymerase uses its template.

### 3.5. Position and Influence of N_tail_ within the Nucleocapsid

N_tail_ belongs to the group of pre-molten globules within the class of intrinsically disordered proteins [34,57,59]. Although N_core_ is sufficient to generate the helical core of the nucleocapsid [35,36,38], N_tail_ influences the conformation of the helix. Indeed, treatment of the nucleocapsid or NCLC by trypsin cleaves off N_tail_, reduces the pitch, and tightens the helix [21,34,113,117,121] (Figure 3B). Moreover, while the expression of the full-length N protein of MuV generates NCLCs in a ring conformation, the cleavage of N_tail_ generates helical structures [126].

In agreement with its disordered nature, N_tail_ is not visible by electron microscopy [34,113,126]. Nuclear magnetic resonance spectroscopy and small-angle scattering showed that N_tail_ escapes from the nucleocapsid core toward the exterior between two helical turns [152] (Figure 7A,B), in agreement with the location of epitopes recognized by antibodies targeting N_tail_ [153]. While the first 50 residues of N_tail_ are spatially constrained by the helical turns of the nucleocapsid core, the C-terminal residues have high angular freedom [75,127,152]. This model is further supported by the recently solved structure of the NCLCs formed by SeV N proteins [21]. After cleavage of most of N_tail_, the nucleocapsids were found to become more rigid and allowed a resolution of 2.9 Å. The atomic model was reconstituted up to residue 414, thus including the first thirteen residues of the tail (aa 402-414). The residues bind the surface of N_CTD_ and point toward the exterior of the nucleocapsid core (Figure 7A).

N_tail_ is thought to mediate the contact between the nucleocapsid and its environment. While N_tail_ is not required for the encapsidation of the RNA, it is essential for RNA synthesis [37,154]. Indeed, N_tail_ contains an α-helical molecular recognition element (α-MoRE) which folds upon binding to the C-terminal domain of the phosphoprotein (P_XD_) and thus participates in the recruitment of the polymerase complex on its template [39,40,71,73,74,75,127,128,153,155,156,157] (Figure 7C). In addition to its interaction with P_XD_, N_tail_ also binds the matrix protein, thus promoting the packaging of the nucleocapsids into viral particles [158,159,160,161,162]. N_tail_ also mediates the interaction with several cellular factors [61], including heat shock protein 70 (hsp70) [154,163,164] and peroxiredoxin 1 [165].

## 4. Formation of the Nucleocapsid

### 4.1. Structure of the N^0^-P Complex

The N-terminal domain of the phosphoprotein interacts with the monomeric, RNA-free form of the N protein (referred to as N^0^), thus forming the so-called N^0^-P complex [55,166]. This interaction prevents the formation of NCLCs on cellular RNAs and recruits the N proteins to the encapsidation site where N^0^-P complexes are used as the substrate for the encapsidation of genomes and antigenomes [7,55,56].

The first 50 residues of P (herein referred to as P_NTD_) of paramyxoviruses contain a conserved α-MoRE [57,59,62,167] that adopts an alpha-helical structure upon binding to N_core_ (Figure 8). The structures of the N^0^-P_NTD_ complex of NiV [24], MeV [25], PIV5 [26], and hPIV3 [27] were solved by X-ray crystallography (Figure 8A–D). To crystallize the complex in a monomeric and RNA-free state, the N_NTDarm_ and N_CTDarm_ were removed (with the exception of MeV in which the N_CTDarm_ was included). For all three viruses, P_NTD_ binds to N_CTD_ in the groove occupied by the N_NTDarm_ of the adjacent N_i+1_ protomer in the nucleocapsid conformation (Figure 8B). The interactions between the groove and the P_NTD_ or the N_NTDarm_ are both of hydrophobic nature. For MeV and NiV, the first alpha-helix is followed by a second alpha-helix segment that binds the top of the N_CTD_ in place of the N_CTDarm_ of the adjacent N_i-1_ protomer (Figure 8B). Contrary to MeV, hPIV3, and NiV, the P_NTD_ of PIV5 extends on its N-terminus and reaches the RNA cavity, potentially directly inhibiting binding to the RNA (Figure 8C). Inhibition of the binding of the arms of the adjacent N protomers by P_NTD_ is a conserved feature among mononegaviruses [131,137,168,169,170,171,172].

For MeV, NiV, and PIV5, the structures of both the RNA-free N_core_ in complex with P_NTD_ and the RNA-bound N_core_ engaged in an NCLC (N^NUC^) have been solved. The overall architecture of N_core_ is conserved between the N^0^ conformation (or “open” conformation) and the N^NUC^ conformation (or “closed” conformation) (Figure 8D). Although individually the N_NTD_ and N_CTD_ have very similar structures, their relative orientation differs (Figure 8D–F). Indeed, the kinked alpha-helix located between the N_NTD_ and the N_CTD_ acts as a hinge and allows the rotation of the N_CTD_ (Figure 8F). This rotation finalizes the formation of the RNA cavity. While NiV P_NTD_ has been suggested to maintain the N protein in the N^0^ conformation by rigidifying the structure [24], modeling and molecular simulations suggest that the PIV5 N protein preferably adopts the N^0^ conformation in the absence of ligand (P_NTD_ or RNA) [26]. The interactions between the RNA backbone and residues from both the N_NTD_ and N_CTD_ most likely participate in the stabilization of the N^NUC^ conformation, even in the absence of adjacent protomers.

In addition to P_NTD_, another ultra-weak interaction was detected between N_core_ and a fragment of the long, disordered region of P located between P_NTD_ and P_OD_ [173]. This fragment contains a short peptide (δ) with no apparent secondary structure and a transient short alpha-helix (α4), which both interact with the bottom of N_NTD_ (Figure 8G). The sequence of α4 has some level of conservation among morbilliviruses and henipaviruses, and the introduction of mutations in this sequence inhibits gene expression in a minigenome assay. The homologous sequence in the NiV P protein also adopts a transient alpha-helix structure in solution [62,174]. Thus, this ultra-weak interaction may be conserved and essential for maintaining the N protein in the N^0^ state [173]. The interaction between the α-MoRE of N_tail_ (N_α-MoRE_) and P_XD_ may also further stabilize the N^0^-P complex [175].

### 4.2. Encapsidation of the RNA

The fusion of full-length N protein (or N_core_) to a cleavable P_NTD_ allows the production of soluble N^0^-P complexes [20,23,176]. Incubation of these complexes with RNA triggers the formation of NCLCs in vitro. This experimental setup revealed that the encapsidation of the RNA is sequence-specific and that the N protein of MeV preferably encapsidates a poly(A) hexamer (but not poly(U)), as well as an RNA corresponding to the first six nucleotides of the genome. Similar specificity was observed for other mononegaviruses, suggesting that the N protein has a higher affinity for poly(A) RNA and the first nucleotides of the corresponding genome [177,178,179,180]. The formation of long helical NCLCs from short RNA hexamers indicates that a continuous RNA molecule is not required for the assembly of N proteins. The kinetics of the encapsidation was analyzed by real-time nuclear magnetic resonance and fluorescence spectroscopy and revealed a two-step mechanism. The first step is rapid and could correspond to the binding of N monomers to the RNA, while the second step consists of the assembly of RNA-bound N proteins [176].

Finally, the local concentration of N^0^-P complexes seems critical to efficient encapsidation of the RNA and generation of NCLCs. Indeed, the formation of droplets formed by liquid-liquid phase separation increases both the local concentration of N^0^-P complexes and the rate of formation of NCLCs in vitro [181]. For VSIV (*Rhabdoviridae* family), although dispensable [182], P_OD_ enhances genome replication possibly by increasing the local concentration of N^0^ at the site of encapsidation [183].

## 5. The Nucleocapsid as a Template

Unlike the distantly related members of the *Articulavirales* and *Bunyavirales* orders for which the polymerase can directly bind the extremity of the viral genomes, the viral polymerase of mononegaviruses cannot interact with its nucleocapsid template without its cofactor P [43]. Moreover, the genomes of mononegaviruses are deeply encased in helical nucleocapsids, thus raising the question of how the polymerase accesses the RNA.

### 5.1. Recruitment and Progression of the Polymerase Complex

Since P_XD_ mediates the interaction of L with the nucleocapsid and the polymerase complex is made of the L protein and a tetramer of P proteins, the polymerase complex possesses four potential nucleocapsid-binding sites. This architecture is the basis of the so-called “cartwheeling” model in which the polymerase complex progresses on its template due to cycles of association/dissociation between P_XD_ and the nucleocapsid [184,185].

The P_XD_ of paramyxoviruses adopts a three-helix bundle structure [71,72,73,75,186,187]. P_XD_ binds to the α-MoRE of N_tail_ [73,74,75,127,128], the L protein [28,188,189], and N_core_ [68,114,190,191] (Figure 9). P_XD_ has a prism-like shape with three faces: α1 + α2 form face 1 (F1), α2 + α3 form face 2 (F2), and α3 + α1 form face 3 (F3) (Figure 9A). Each interaction is mediated by a different face, with N_α-MoRE_ interacting with F2 [73], the L protein with F3 [28,189], and N_core_ with F1 [191] (Figure 9).

The interaction of P_XD_ with N_α-MoRE_ is weak, with a constant of dissociation (K_D_) in the µM range [74,128,157,193]. While the disordered N_α-MoRE_ folds into an alpha-helix upon binding to P_XD_, the flanking regions of N_tail_ remain disordered [74,75,127,187,194,195,196,197]. This phenomenon, called fuzziness [198], affects the interaction between N_tail_ and its partners. Indeed, the long-disordered region upstream of N_α-MoRE_ slows the disorder-to-order transition of N_α-MoRE_ upon binding to P_XD_ [199,200]. Accordingly, the deletion of the disordered region upstream MeV, NiV, and HeV N_α-MoRE_ increases the affinity for P_XD_ (and also HSP70 for MeV) [199]. The dampening effect of the disordered regions does not directly depend on the sequence, but rather on a combination of length and disorder. Although not directly involved, the poorly conserved disordered region of N_tail_ upstream the N_α-MoRE_ affects the interaction with P_XD_ [199,200,201].

The low affinity between P_XD_ and N_α-MoRE_ permits efficient progression of the polymerase complex on the nucleocapsid. Indeed, modifications of the binding strength between MeV P_XD_ and N_α-MoRE_ alter the activity of the polymerase [202,203,204]. Indeed, there is a positive correlation between the P_XD_/N_α-MoRE_ interaction strength and the efficiency of the re-initiation of the transcription at gene junctions [204] (Figure 2A). This effect is further enhanced when the polymerase crosses long non-transcribed intergenic regions, suggesting the P_XD_/N_α-MoRE_ interaction keeps the polymerase in an active state when crossing the gap between transcription units.

Although the P_XD_/N_α-MoRE_ interaction regulates the activity of the polymerase, it is not strictly required for RNA synthesis. Indeed, while the truncation of MeV N from N_α-MoRE_ to its C-terminus abrogates gene expression in a minigenome assay, further deletion of the 43 upstream residues restores activity [205] (Figure 10). This suggests that the central disordered region (N_CDR_) upstream of the N_α-MoRE_ has a negative effect on the activity of the polymerase and the binding of P_XD_ to N_α-MoRE_ counteracts this effect. Accordingly, deletions in the N_CDR_ of CDV and NiV increase activity [206], and when MeV N_α-MoRE_ is relocated to a variable region of N_NTD_, removing N_CDR_ enhances gene expression [207] (Figure 10). However, a recombinant MeV harboring N_α-MoRE_ on N_core_ and devoid of N_CDR_ produces more polycistronic mRNAs. Similarly, two recombinant CDV with deletions in N_CDR_ produce more polycistronic mRNA and show an altered RNA synthesis balance with enhanced transcription over replication [206]. Therefore, the central disordered region of N_tail_ inhibits RNA synthesis, and P_XD_ counterbalances this negative effect. This interplay would finely tune the polymerase’s activity and ensure the quality of the generated mRNAs by improving the recognition of the signals at the intergenic junctions.

In addition to interacting with the α-MoRE, P_XD_ of CDV was also shown to directly interact with N_core_ [191]. This interaction is mediated by an acidic loop of N_NTD_ (residues 146 to 161) and requires the presence of N_α-MoRE_ in complex with P_XD_ (Figure 9C–E). Accordingly, mutations in the homologous region of NiV N_core_ were previously shown to inhibit the P-N interaction [192]. After several passages of a recombinant CDV expressing a N_tail_ with a large deletion in N_CDR_, compensatory mutations appeared in P_XD_ and N_NTD_ that decreased the affinity between the two domains. Thus, N_CDR_ may inhibit the interaction of P_XD_ with N_core_. Modeling and molecular docking predict that the P_XD_/N_α-MoRE_ complex binds between two N protomers. On one side the F1 of P_XD_ binds to the acidic loop of one protomer and on the other side, N_α-MoRE_ interacts with a different loop (residues 133 to 142) of the N_NTD_ of the other protomer (Figure 9D,E). The requirement of N_α-MoRE_ for this interaction suggests P_XD_ first binds N_α-MoRE_ and then N_core_. A similar interaction with the acidic loop may explain the singular behavior of MuV P_XD_, which can bind N_core_ in the absence of N_tail_ [68,190]. This atypical phenotype may be linked to the lower stability of the MuV P_XD_ tertiary structure in solution and to a folding-upon-binding mechanism [186,190]. The binding of CDV P_XD_ to N_NTD_ also agrees well with the density map of MuV nucleocapsids bound to P_XD_ [114]. The P protein of the respiratory syncytial virus (*Pneumoviridae* family) also binds the nucleocapsid on the same location on N_NTD_, suggesting an ancestral mode of interaction between the polymerase complex and its template [191,208,209].

P binds the L protein via the C-terminal part of the P_OD_ and the third face of the P_XD_ of one P protomer [28,188,189] (Figure 9F). Using different combinations of mutants in a minigenome assay, it was shown that at least one P_XD_ of the tetramer must have the ability to bind the L protein and one different P_XD_ must be able to bind N_α-MoRE_ [189]. The P_XD_ that binds L does not have to be able to bind N_α-MoRE_ and it is unclear whether P_XD_ can simultaneously bind L and N_α-MoRE_. It is also unknown whether the interaction between L and P_XD_ is stable or undergoes rounds of association/dissociation during RNA synthesis.

Of note, some cellular factors influence the progression of the polymerase complex. For instance, hsp70 [163] and peroxiredoxin 1 [165] both compete with P_XD_ for the binding to N_tail_ and enhance RNA synthesis.

Taken together, these results show that P_XD_ is at the core of an interaction network, which is more complex than initially anticipated. P_XD_ participates in a finely regulated dance allowing the progression of the polymerase complex onto the nucleocapsid and the optimal recognition of the transcription signals.

### 5.2. Access to the RNA

The viral genome is encased between the N_NTD_ and N_CTD_ of each N protomers, with only half of its bases facing the solvent. Thus, copying of the genome requires significant conformational rearrangements to allow the RNA to reach the catalytic site of the polymerase.

First, in the straight helical conformation of purified nucleocapsids, the RNA is covered by the N protomers of the adjacent helical turn, leaving an insufficient empty space above the RNA to accommodate the large polymerase complex (Figure 5C). Therefore, a local modification of the conformation of the helical nucleocapsid must take place. Nucleocapsids are flexible and can adopt different conformations [21,109,110,111]. Such uncoiling and bending could enable the polymerase complex to reach the RNA cavity. Although N_tail_ affects the rigidity and the pitch of the helical nucleocapsids [21,113,117,121], the addition of monomeric P_XD_ to intact nucleocapsids of HeV or MuV does not affect the coiling of the helix [75,114]. For HeV, the addition of P_XD_ does not make the RNA more accessible to nucleases either. Similarly, the binding of the C-terminal domain of VSIV P to NCLC does not significantly modify the structure of the N proteins [210]. This suggests that either the binding of P_XD_ does not participate in the uncoiling of the nucleocapsid, or the oligomerization of P_XD_ is required. Some cellular factors may also contribute to the uncoiling of the nucleocapsid, such as hsp70, which binds MeV N_tail_ [154,163], stimulates MeV and CDV polymerase activity [154,211], and seems to favor the production of “light” nucleocapsids (versus “dense” nucleocapsids) of CDV [212,213]. Conversely, the P_NTD_ of MuV can uncoil the helical nucleocapsid and increase gene expression in a minigenome assay [68,114]. Similar to the binding of MuV P_XD_ to N_core_, the atypical behavior of MuV P_NTD_ could reflect a possibly common uncoiling mechanism.

Second, the RNA cavity of the N proteins must open to release the RNA upon the passage of the polymerase. A first model proposes that the N protomers undergo a large conformational rearrangement reverting from the N^NUC^ to the N^0^ conformation [24,214]. Molecular dynamics simulations show that in absence of ligands (RNA or P_NTD_), the N protein of PIV5, initially in the N^NUC^ conformation, adopts a conformation similar to the N^0^ conformation. This suggests that the N^0^ conformation is the most stable in the absence of RNA and adjacent N proteins [26]. However, the removal of the RNA from nucleocapsid-like structures of MuV or VSIV does not significantly change the relative orientation of N_NTD_ and N_CTD_ [179,215]. Moreover, when the simulations are made on a short RNA-free nucleocapsid of three N proteins of PIV5, only the conformations of the N_i+1_ and N_i_ protomers slightly change toward the N^0^ conformation, with the conformation of the N_i_ protomer being modified to a lower extent. Thus, the conformational rearrangement from the N^NUC^ to the N^0^ conformation may partially take place only at the 3′ end of the genome to facilitate the release of the RNA terminus and the engagement of the polymerase.

Another model proposes that a local conformational change of the α-helix 7 and the upstream flexible loop is enough to release the RNA [215] (Figure 8E). Indeed, this helix, which is located at the surface of the protein, constitutes most of the bottom part of the RNA cavity and contains several amino acids implicated in RNA stabilization. Moreover, when the N_NTD_ of MeV N^NUC^ and N^0^ are superimposed, the loop-α7 region is the only fragment that does not properly align (Figure 8E). Thus, in addition to the rotation of the N_NTD_, the transition from the N^0^ to the N^NUC^ conformation also implies a local rearrangement of this region. Mutations in this helix affect gene expression, which is in agreement with a potential role in releasing the RNA during RNA synthesis [215].

P_NTD_ has also been proposed to participate in the release of the RNA. Due to the tight N-N interactions and the high stability of the nucleocapsid, it seems unlikely that during the progression of the polymerase P_NTD_ would bind the N protomers in an N^0^-P-like manner and sequentially “open” the N proteins. However, the N_NTDarm_-binding groove of the last N protomer of the nucleocapsid (or the first protomer on the 3′ terminus) is not occupied and may therefore be engaged by P_NTD_ to help to release the RNA. Due to the binding register of the first six nucleotides of the genome and the rule of six, the terminal 3′ nucleotides are positioned such that they are exposed to the solvent and are therefore more accessible [20]. Therefore, the binding of P_NTD_ only to the terminal N protomer may be sufficient for the polymerase to access the RNA. However, for SeV and hPIV3, the deletion of the P_NTD_ abolishes genome replication but not transcription in vitro, thus suggesting that the potential role of P_NTD_ in releasing the RNA is not essential (at least for members of the *Respirovirus* genus) [216,217].

## 6. Model of Transcription and Replication

### 6.1. Transcription

Upon fusion of viral and cellular membranes and the subsequent release of the nucleocapsid in the cytoplasm, the polymerase complexes previously packaged in the virion initiate primary transcription. Given that the nucleocapsid adopts a helical conformation, the two elements of the bipartite promoter are exposed on the same side. P_XD_ binds N_tail_ and recruits the polymerase complex on the nucleocapsid (Figure 11A). When the polymerase reaches the 3′ end of the nucleocapsid, it is somehow stabilized by the second element of the bi-partite promoter (PE2). The phase register of the encapsidation exposes the 3′ end of the genome to the solvent. The release of the RNA from the first N protein may be enhanced by the lodging of one P_NTD_ into the empty N_NTDarm_-binding groove of the first protomer (Figure 11B). A conformational change of the first two N protomers from the N^NUC^ to the N^0^ conformation may also assist the release. When the polymerase complex passes on an N protomer, a transient local conformational change of the helix α7 of N_NTD_ and the preceding loop allows the release of the RNA (Figure 11C).

The progression of the polymerase along the genome induces a local conformational change of the helical nucleocapsid (Figure 11D). A dynamic dance is orchestrated by P_XD_, which periodically binds the α-MoRE of N_tail_ and N_core_. The fuzziness of N_tail_ inhibits the passage of the polymerase complex, which uses P_XD_ to “clear the way”. First, P_XD_ binds N_α-MoRE_ which folds upon binding into an alpha-helix (Figure 11E). Second, the P_XD_/N_α-MoRE_ complex binds the N_core_ of the same protomer or an adjacent protomer (model 1), thus stabilizing the N_tail_, decreasing the fuzziness, and facilitating the progression of the polymerase (Figure 11F,G). As the binding of the P_XD_/N_α-MoRE_ complex to the N_NTD_ of N protomer from the same helical turn would potentially further block the way (Figure 11F), the complex could alternatively bind the N_NTD_ of a protomer from the previous helical turn (model 2). In this alternative model, the anchoring of the N_tail_ to the previous helical turn would further remove it from the path of the polymerase (Figure 11H–K).

This dynamic interaction network regulates the speed of the RNA synthesis and ensures an optimal recognition of the transcription signals at the intergenic regions (IR) (Figure 2A). Indeed, upon the recognition of the first IR, the polymerase stops the synthesis of the leader RNA and scans the genome to find the next gene start signal (GS). Recognition of the GS triggers the re-initiation of RNA synthesis and capping of the mRNA. If recognized, the gene end signal triggers the synthesis of the poly(A) tail (for more details on the mechanisms of RNA synthesis, see [218]). The efficiency with which the polymerase crosses the IR and scans the genome depends on P_XD_ and its interactions with the nucleocapsid. It is yet to be deciphered if the P_XD_ that binds the L protein can simultaneously bind N_tail_ and if it periodically associates with and dissociates from L.

### 6.2. Replication

After new viral proteins have been synthesized, the polymerase complex switches from a transcriptase mode to a replicase mode. The details of this switch are not fully understood, but the accumulation of the encapsidation substrate, the N^0^-P complex, plays a key role in this process. In the replicase mode, the polymerase complex encapsidates the nascent RNA and does not recognize any transcription signals. As the N protein has a higher affinity for an RNA with the sequence of the 5′ end of the genome, the first six nucleotides of the nascent genome outcompete P_NTD_ for the binding to N (Figure 11L,M). For the following N protomers, P_NTD_ is outcompeted by both the RNA and the N_CTDarm_ of the previous N protomer (Figure 11N). This double competition allows the continuous encapsidation of any RNA sequences, even with a poor affinity for N, such as poly(U) sequences.

## 7. Conclusions and Perspectives

In recent years, several significant advances have been made which add to our understanding of the structure and mechanisms of the molecular machinery underlying RNA synthesis of paramyxoviruses. We now have in hand structural information on the N^0^-P complex and the nucleocapsid of three and seven viruses, respectively. The determination of the phase register of the encapsidated RNA further reveals the mechanisms underlying the rule of six. Moreover, the deciphering of the rich interaction network connecting the different components of the ribonucleoprotein complex gives us a good overview of the dynamics of the progression of the polymerase complex. Finally, the resolution of the atomic structure of the polymerase complex of PIV5 provides a new tool and opens new opportunities to further elucidate the molecular details of the RNA synthesis.

Except for MuV and its atypical features (antiparallel coiled-coil for P_OD_, unstable P_XD_ in solution, interaction between P_XD_ and N_core_ in the absence of N_tail_, and uncoiling of the nucleocapsid by P_NTD_), the molecular machinery and the mechanisms at play during RNA synthesis appear to be well conserved among paramyxoviruses. The exposure of the RNA on the outside of the helical nucleocapsid, the binding site of the C-terminal domain of P to the N_NTD_, the coiled-coil structure of P_OD_, and the interaction sites between P and L suggest these mechanisms are shared with pneumoviruses and filoviruses but are quite different for rhabdoviruses.

Despite these new insights, several questions remain unanswered. The helical nucleocapsid must uncoil to allow the progression of the polymerase. What factors induce this conformational change? It is also still unclear how the nucleoproteins release the RNA and what signals trigger this release. Although structural information is available on the individual components, there is little insight into the organization or the structure of the polymerase complex on the nucleocapsid while synthesizing RNA. In addition, how does the second element of the bipartite promoter enhance initiation? How does the polymerase complex add nucleoproteins to the nascent RNA? How does the phosphoprotein orchestrate its interactions with N_tail_, N_core_, and L? How do other viral proteins such as the V, C, and M proteins interact with the ribonucleoprotein complex and regulate RNA synthesis?

Answering these questions should bring significant insight into the mechanisms at play and assist the development of safe and potent vaccine vectors, oncolytic viruses, and antiviral therapeutics. With the momentum provided by the recent advances and the constant improvement of powerful tools such as cryo-electron microscopy, there is no doubt our understanding of the viral molecular machinery of paramyxoviruses will continue to rapidly improve.

## Figures and Tables

**Figure 1 viruses-13-02465-f001:**
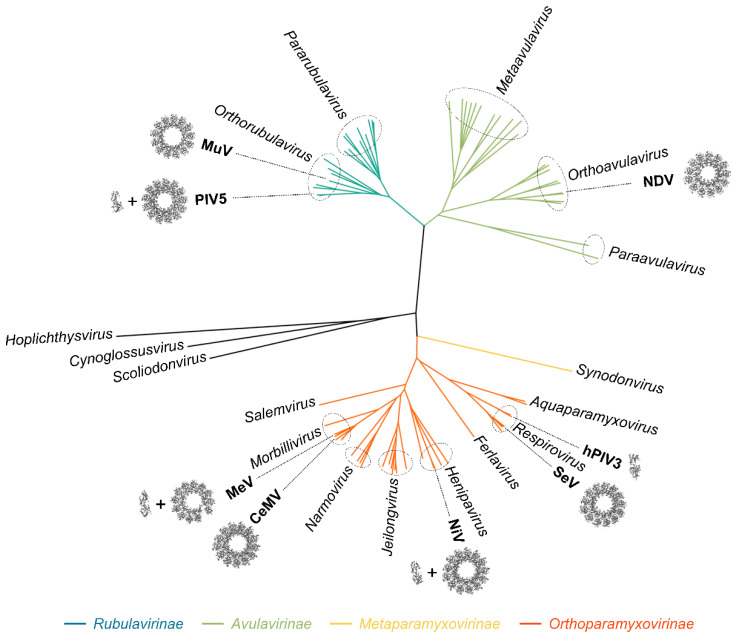
Phylogenetic tree of the *Paramyxoviridae* family. The phylogenetic tree was generated from an alignment of full-length L proteins. One sequence per species was used. The 78 sequences were selected based on the GenBank accession numbers given by the 2020 taxonomy of the International Committee on Taxonomy of Viruses. The names of the genera are indicated in italics. In bold, viruses for which structural data on the N protein is available. Structures of rings or single helical turns of the nucleocapsid-like complexes are shown from a top view (Newcastle disease virus (NDV), reconstructed from PDB: 6JC3; Sendai virus (SeV), reconstructed from PDB: 6M7D; Nipah virus (NiV), PDB: 7NT5; cetacean morbillivirus (CeMV), PDB: 7OI3; measles virus (MeV), PDB: 6h5Q; parainfluenza virus 5 (PIV5), PDB: 4XJN; mumps virus (MuV), PDB: 7EWQ). For human parainfluenza 3 (hPIV3), PIV5, MeV, and NiV, the structures of the N^0^-P complexes are shown (hPIV3, PDB: 7EV8; PIV5, PDB: 5WKN; MeV, PDB: 5E4V; NiV, PDB: 4CO6).

**Figure 2 viruses-13-02465-f002:**
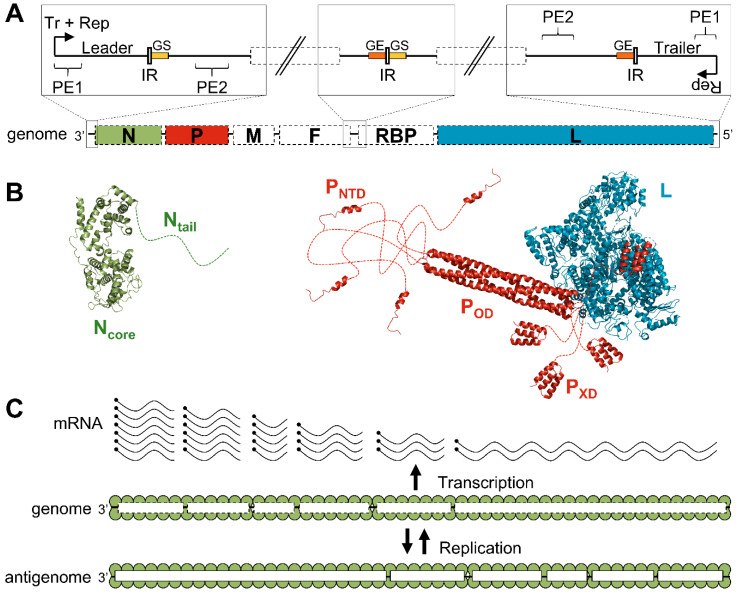
Organization and structure of the components of the RNA synthesis machinery. (**A**) Schematic representation of the viral genome and the regulatory elements. The genome contains at least six conserved adjacent genes separated by intergenic regions (IR). These genes encode for the nucleoprotein (N), the phosphoprotein (P), the matrix protein (M), the fusion protein (F), the receptor-binding protein (RBP), and the large protein (L). The promoters located at the 3′ end of the genome (transcription and replication) and the antigenome (replication only) are bipartite and made of two promoter elements (PE1 and PE2). The transcription of each gene starts on a “gene start” signal (GS, in yellow) and ends on a “gene end” signal (GE, orange). (**B**) Cartoon representation of the structure of the protein components of the RNA synthesis machinery of PIV5 (N, PDB: 4XJN; P_NTD_, PDB: 5WKN; P_OD_, P_XD_, and L, PDB: 6V85). Disordered regions are represented as dotted lines. (**C**) Schematic representation of viral transcription and replication. Encapsidated genomes are transcribed into a gradient of mRNAs. Replication of the genomes requires the production of encapsidated antigenome intermediates.

**Figure 3 viruses-13-02465-f003:**
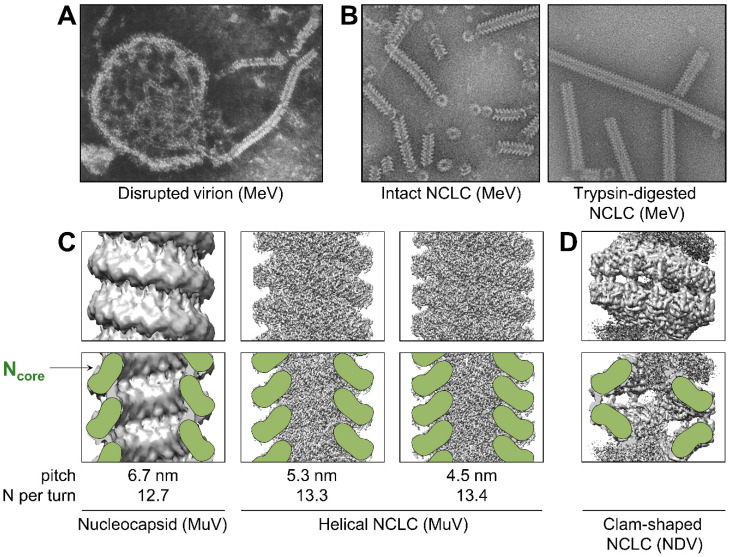
Conformations of nucleocapsids. (**A**) Disrupted measles particle by negative-stain microscopy. Republished with permission of Journal of Virology, from [116]; permission conveyed through Copyright Clearance Center, Inc. (**B**) Intact (left) and trypsin-digested (right) nucleocapsid-like complexes (NCLC) of MeV. Republished with permission of Journal of Virology, from [117]; permission conveyed through Copyright Clearance Center, Inc. (**C**) Top row: density maps of MuV nucleocapsid (left, EMDB: EMD-2630) and MuV NCLC (center, EMBD: EMD-31368; right, EMBD: EMD-31369). Bottom row: cross sections of the density maps shown above with schematic representation of N_core_ in green. The pitch and number of N protomers per helical turn are indicated at the bottom. (**D**) Top: density map of a clam-shaped structure of NDV (EMBD: EMD-9793). Bottom: cross section of the density map shown above with schematic representation of N_core_ in green.

**Figure 4 viruses-13-02465-f004:**
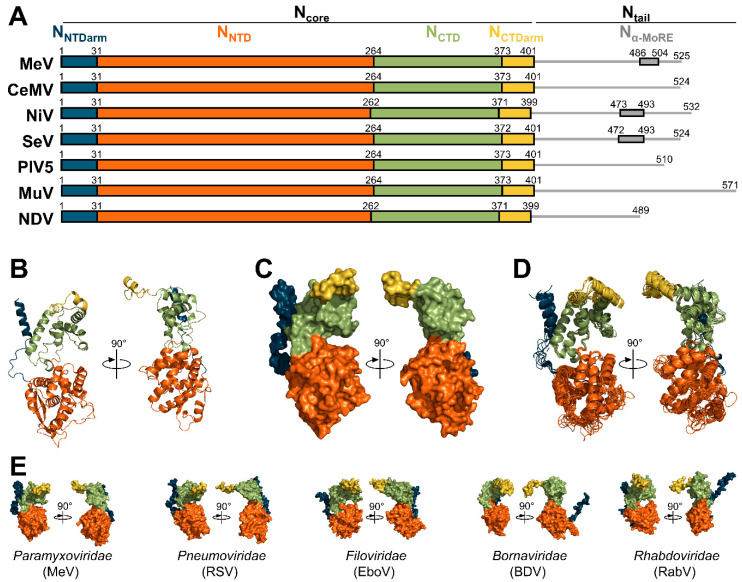
Organization and structure of the nucleoproteins. (**A**) Organization and boundaries of the domains of all nucleoproteins whose structure has been solved. Boundaries of N_α-MoRE_ are only known for MeV [71,73], NiV [74,127], and SeV [128]. (**B**) Structure of the N protein of MeV in cartoon representations (PDB: 6H5Q). (**C**) Structure of the N protein of MeV in surface representations (PDB: 6H5Q). (**D**) Superimposition of aligned structures of MeV (PDB: 6H5Q), CeMV (PDB: 7OI3), NiV (PDB: 7NT5), SeV (PDB: 6M7D), PIV5 (PDB: 4XJN), MuV (PDB: 7EWQ), and NDV (PDB: 6JC3). (**E**) Surface representations of the structure of the N proteins of MeV (PDB: 6H5Q), respiratory syncytial virus (RSV, PDB: 2WJ8), Ebola virus (EboV, PDB: 5Z9W), Borna disease virus (BDV, PDB: 1N93), and rabies virus (RabV, PDB: 2GTT). All the structures shown in Figure 4 correspond to N proteins observed in their assembled form (i.e., in NCLC).

**Figure 5 viruses-13-02465-f005:**
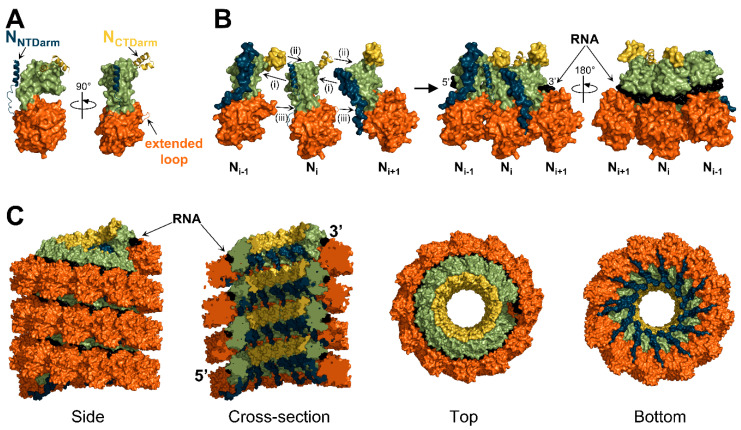
Mode of assembly. (**A**) Structure of the nucleoprotein of PIV5 shown in surface representation, except for the N_NTDarm_, N_CTDarm_, and extended loop, which are shown in cartoon representation (PDB: 4XJN). (**B**) Surface representation of three N protomers with the N_NTDarm_, N_CTDarm_, and extended loop of the N_i_ protomer shown in cartoon representation. The RNA is shown in black. (**C**) Surface representation of a nucleocapsid-like complex of MeV shown with four different views (reconstituted from PDB: 6H5Q). Color code is the same as in Figure 4.

**Figure 6 viruses-13-02465-f006:**
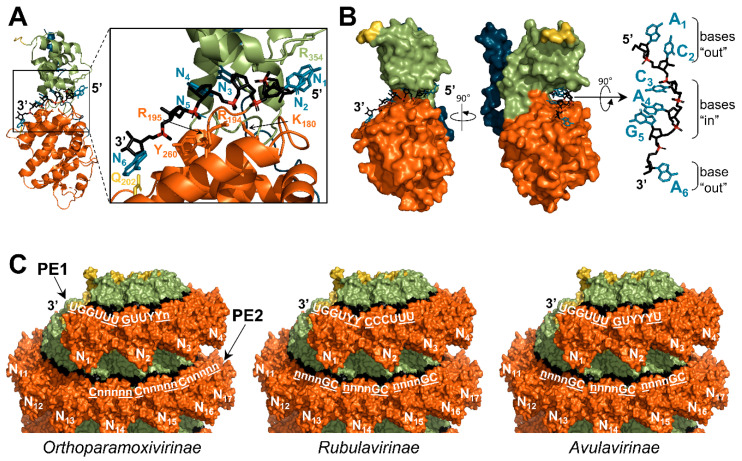
Position of the RNA. (**A**) Cartoon representation of MeV nucleoprotein bound to the first six nucleotides of the genome (5′ end) (PDB: 6H5S). The RNA and the residues implicated in the binding to the RNA are represented with sticks. The bases of the RNA are shown in blue, the backbone in black, and the phosphates in red. Residue Q202 is shown in yellow. (**B**) Same as (**A**) with the nucleoprotein in surface representation. (**C**) NCLCs of CeMV are shown in surface representation (reconstitution of the helix from PDB: 7OI3). For each subfamily, the consensus sequences of the promoter elements (PE1 and PE2) are shown in white. Well-conserved nucleotides are in capital letters. Bases in “out” positions are underlined. For N, same color code as in Figure 4.

**Figure 7 viruses-13-02465-f007:**
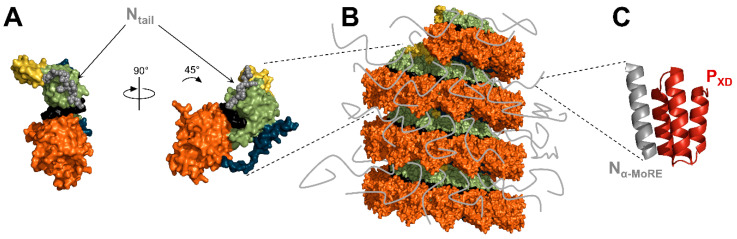
Structure and position of N_tail_. (**A**) Structure of the nucleoprotein of SeV in surface representation with the residues of N_tail_ shown with spheres (PDB: 6M7D). (**B**) Surface representation of the nucleocapsid-like complex of CeMV with the N_tail_ represented as grey lines (reconstitution from PDB: 7OI3). (**C**) Structure of MeV P_XD_ in complex with N_α-MoRE_ (PDB: 1T6O). For N, the color code is the same as in Figure 4.

**Figure 8 viruses-13-02465-f008:**
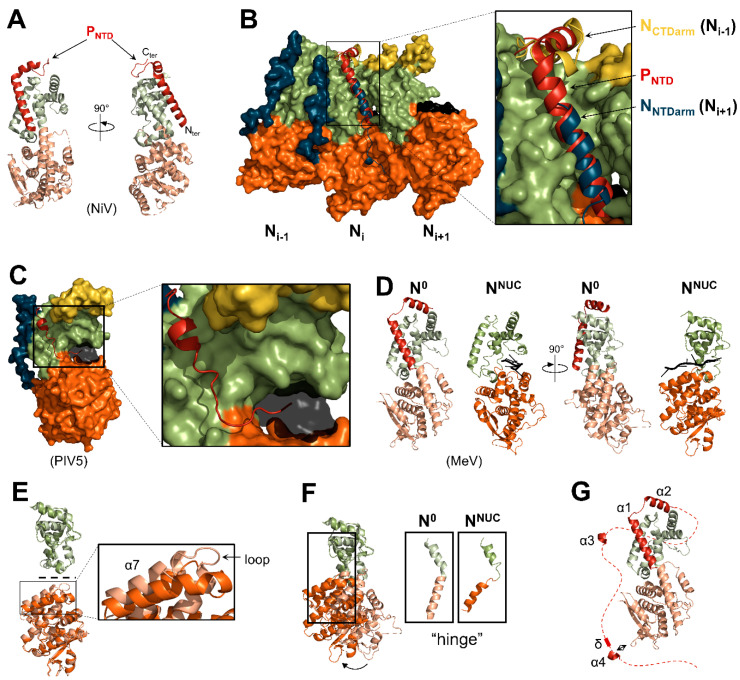
Structure of the N^0^-P complex. (**A**) Structure of the N^0^-P complex of NiV (PDB: 4CO6). The N_NTD_ and the N_CTD_ are in pale orange and pale green, respectively. (**B**) Superimposition of the structure of three N protomers of NiV NLPC (PDB: 7NT5) with NiV N^0^-P (PDB: 4CO6). The N_CTD_ of the N^0^-P complex was aligned onto the N_CTD_ of the N_i_ promoter. The N protomers are presented in surface representation, while the N_NTDarm_, N_CTDarm_, and the extended loop of the N_i_ protomer are shown in cartoon representation. P_NTD_ is shown in cartoon representation. (**C**) Structure of the N^0^-P complex of PIV5 with P_NTD_ in cartoon representation (PDB: 5WKN). The RNA is shown as a transparent black surface. (**D**) Side-by-side comparison of the structures of the N^0^-P complex (PDB: 5E4V) and of an N protomer of the NCLC of MeV (PDB: 6H5Q) (N_NTDarm_ and N_CTDarm_ are not shown). The N_NTD_ and the N_CTD_ of N^0^ are in pale orange and pale green, respectively. (**E**) Superimposition of the N_CTD_ (top) and of the N_NTD_ (bottom) of the N^0^-P complex and of an N protomer of the NCLC of MeV. (**F**) Superimposition of the structures of MeV N^0^ and N^NUC^ (N_NTDarm_ and N_CTDarm_ are not shown). The structures are aligned based on their N_CTD_. (**G**) Structure of the MeV N^0^-P complex with the disordered region downstream P_NTD_ shown as a dotted line. The transient alpha-helices (α3 and α4) are shown in cartoon representation.

**Figure 9 viruses-13-02465-f009:**
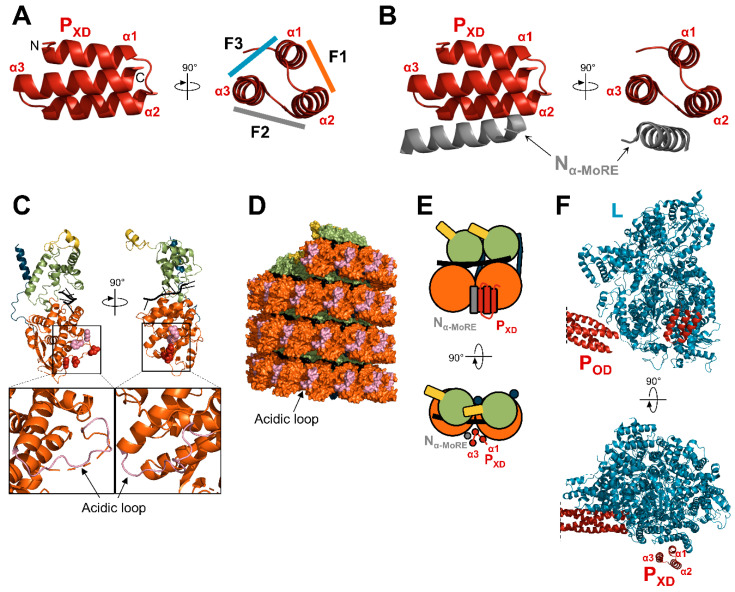
Interaction network of P_XD_. (**A**) Structure of MeV P_XD_ (PDB: 1T6O). The three faces of the “prism” are indicated (F1, F2, F3). F1, F2, and F3 are shown with a color code corresponding to the color used to draw the protein partner. (**B**) Structure of MeV P_XD_ in complex with N_α-MoRE_ (PDB: 1T6O). (**C**) Structure of a protomer of MeV N, as observed in the NCLC (PDB: 6H5Q) with the RNA shown in black. Amino acids shown as red spheres correspond to CDV N residues that once mutated restore the growth of a recombinant CDV bearing deletions in the N_CDR_ [191]. The amino acids corresponding to the residues of NiV identified as important for the interaction with P_XD_ are shown as pink spheres [192]. Bottom: enlargements of the N_NTD_ with the acidic loop shown in pink. (**D**) Surface representation of a nucleocapsid-like complex of MeV (reconstituted from PDB: 6H5Q) with the acidic loop shown in pink. (**E**) Schematic representation of two adjacent N protomers bound to P_XD_ in complex with N_α-MoRE_. (**F**) Structure of the polymerase complex of PIV5 (PDB: 6V85). For panels (**C**–**E**), N protomers are colored according to the color code in Figure 4.

**Figure 10 viruses-13-02465-f010:**
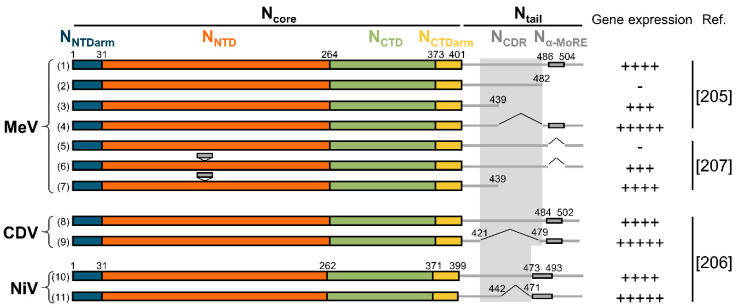
Summary of experimental data on the effect of N_tail_ on the polymerase’s activity. Schematic representation of recombinant N proteins analyzed in minigenome assays. The levels of gene expression are normalized to the level of the wild-type protein (first protein of each set). Activity levels: 0–10%: “-”; 50–75%: “+++”; 75–100%: “++++”; >100%: “+++++”. For mutants (6) and (7) N_α-MoRE_ is inserted in a loop of N_NTD_ at residue 138. Mutants (4), (5), (6), (9), and (11) contain internal deletion shown with a “^”. Activity levels are inferred from the references indicated on the right.

**Figure 11 viruses-13-02465-f011:**
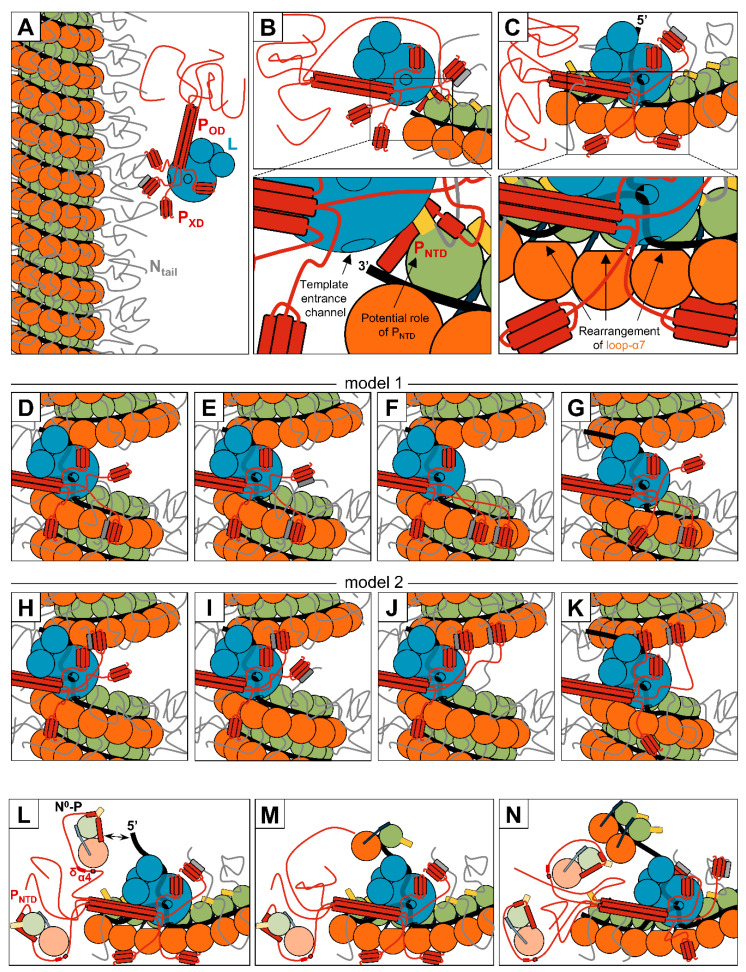
Models of different steps of RNA synthesis. Schematic representations of the recruitment of the polymerase complex (**A**), the initiation of RNA synthesis (**B**), the release of the RNA (**C**), the dynamics of the interaction between P_XD_ and N_tail_ and of P_XD_-N_α-MoRE_ with N_core_ (**D**–**K**), and the encapsidation of nascent RNA during replication (**L**–**N**). (**A**) The polymerase complex is recruited on its template via the interaction between P_XD_ and N_tail_. (**B**) The polymerase complex finds the 3′ end of the genome to initiate RNA synthesis. Bottom panel: enlargement of the top panel. The binding of P_NTD_ to the first protomer may enhance the release of the RNA and its engagement in the template entrance channel. (**C**) The progression of the polymerase complex induces a local rearrangement of the helix α7 and the preceding loop. This conformational change releases the RNA. (**D**–**K**) Two models are proposed for the dynamics of the interaction between P_XD_ and N during the progression of the polymerase complex. (**D**,**H**) In both cases, the polymerase induces a local conformational change on the helical nucleocapsid. The N_tail_ are flexible, disordered, and inhibit the progression of the polymerase complex. (**E**,**I**) P_XD_ binds N_tail_ and induces the folding of the N_α-MoRE_. (**F**) In model 1, the P_XD_/N_α-MoRE_ complex binds the N_core_ of the next N protomer. (**G**) The stabilization of N_tail_ by P_XD_ allows the polymerase complex to continue RNA synthesis. (**J**) In model 2, the P_XD_/N_α-MoRE_ complex binds the N_core_ of an N protomer from the previous helical turn. (**K**) The anchoring of the N_tail_ to the previous helical turn “clears the way” and allows an efficient progression of the polymerase complex. (**L**) During replication, some P_NTD_ are bound to N^0^ monomers. The N^0^-P complex is stabilized by the transient α4 and the preceding delta domain. N^0^ has a strong affinity for the first six residues of the viral genome. (**M**) N^0^ is transferred onto the nascent RNA and switches from the N^0^ to the N^NUC^ conformation. (**N**) Thanks to interactions between the N^0^ and the last N protomer added on the nascent RNA, the N proteins can be added to the growing polymerase even if the affinity for the RNA sequence is low. For N, the color code is the same as in Figure 4.

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
