# Peer review of "The Nucleocapsid of Paramyxoviruses: Structure and Function of an Encapsidated Template"

_viruses, 2021, doi:10.3390/v13122465_

Round 1

Reviewer 1 Report

The author should be commended for a well structured review of paramyxovirus nucleocapsids. I have only minor suggestions and endorse publication whether or not my suggestions are included. 

Suggestions.

  • Perhaps the title should be reconsidered. While much of the review focuses on the nucleocapsid, as expected the role of both L and P in a biological context are extensively discussed. The authors may wish to expand the title to make reference to "replication machinery" or some such term

  • In figure 1 perhaps include the PDB ID's as was done in the other structural figures.

  • Recently an PIV3 L-protein dimer structure has been reported in the literature (https://doi.org/10.1101/2021.09.13.46008). The authors may wish to comment on how this possible dimer may fit into their scheme in figure 11. This structure (dependant on release of the PDB) may be worth discussing.

  • Recently (perhaps while under review) a PIV3 N-P has been deposited in the PDB ID:7EV8. I am not aware of a publication related to this structure but as the coordinates are released it should be included.
  • A recent work (https://doi.org/10.1101/2021.08.04.455049) has demonstrated native back-to-back ring structures for RSV. This isn't a paramyxovirus it shows that these closed ring structures appear to form in cells. Perhaps this is worth including. 

Author Response

Response to Reviewer 1

The author should be commended for a well structured review of paramyxovirus nucleocapsids. I have only minor suggestions and endorse publication whether or not my suggestions are included. 

Suggestions.

  • Perhaps the title should be reconsidered. While much of the review focuses on the nucleocapsid, as expected the role of both L and P in a biological context are extensively discussed. The authors may wish to expand the title to make reference to "replication machinery" or some such term

Response: I agree that the review covers not only the structure of the nucleocapsid but also the use of the nucleocapsid as a template (as indicated in the current title). However, I do not wish to expand the title since there would be much more to talk about to cover the “replication machinery” of paramyxoviruses (catalytic activities of the polymerase, the domains of the L protein, the role of the oligomerization domain of P…). Therefore, I prefer to keep the title as is.

  • In figure 1 perhaps include the PDB ID's as was done in the other structural figures.

Response: PDB ID’s have been added to the legend.

  • Recently an PIV3 L-protein dimer structure has been reported in the literature (https://doi.org/10.1101/2021.09.13.46008). The authors may wish to comment on how this possible dimer may fit into their scheme in figure 11. This structure (dependant on release of the PDB) may be worth discussing.

Response: I am aware of this work and I agree that the dimerization of the polymerase is an interesting observation. However, as of today, this work has not been published in a peer-reviewed journal (only on BioRxiv) and the PDB files are not accessible. Therefore, I do not wish to include this work in this review.

  • Recently (perhaps while under review) a PIV3 N-P has been deposited in the PDB ID:7EV8. I am not aware of a publication related to this structure but as the coordinates are released it should be included.

Response: As of last month, this work has been published in Journal of Virology. I added the structure of the N0-P complex of hPIV3 in figure 1 and I refer to this work three times in the text (with reference # 27).

  • A recent work (https://doi.org/10.1101/2021.08.04.455049) has demonstrated native back-to-back ring structures for RSV. This isn't a paramyxovirus it shows that these closed ring structures appear to form in cells. Perhaps this is worth including. 

Response: I agree that the formation of rings in infected cells is an interesting observation. However, because this work has not yet been published in a peer-reviewed journal, I decided to not mention it.

Reviewer 2 Report

Please see the atatchment

Author Response

This review is well written and easy to read. In particular, I have found that the figures are very informative for illustrating the three dimensional nucleocapsid organization of paramyxoviruse, mapping the key elements of N and P proteins within the nucleocapsid, and understanding the dynamic structure-function relationship of the nucleocapsid during viral RNA synthesis. The author also provides a balanced review of the original and recent literatures, as well as discusses important unanswered questions in the field. For figure 11, it would be helpful to indicate the color code in the legend (e.g, the same color code as in figure 4).

Response: At the end of the legend of Figure 11, I added: "For N, same color code as in figure 4."